# Cell-free prediction of protein expression costs for growing cells

Olivier Borkowski[1,2], Carlos Bricio[1,2], Michela Murgiano[1,2], Brooke Rothschild-Mancinelli[1,2], Guy-Bart Stan [1,2] & Tom Ellis [1,2]

Translating heterologous proteins places significant burden on host cells, consuming expression resources leading to slower cell growth and productivity. Yet predicting the cost of protein production for any given gene is a major challenge, as multiple processes and factors combine to determine translation efficiency. To enable prediction of the cost of gene expression in bacteria, we describe here a standard cell-free lysate assay that provides a relative measure of resource consumption when a protein coding sequence is expressed. These lysate measurements can then be used with a computational model of translation to predict the in vivo burden placed on growing *E. coli* cells for a variety of proteins of different functions and lengths. Using this approach, we can predict the burden of expressing multi-gene operons of different designs and differentiate between the fraction of burden related to gene expression compared to action of a metabolic pathway.

[1] Imperial College Centre for Synthetic Biology, Imperial College London, London SW7 2AZ, UK. [2] Department of Bioengineering, Imperial College London, London SW7 2AZ, UK. Correspondence and requests for materials should be addressed to T.E. (email: t.ellis@imperial.ac.uk)

To be able to build systems with increasingly more genes, not only is precise gene expression desirable, but it is also essential to have an understanding of the burden these will place on the host cell so that designs can be optimised to ensure robust growth and to prevent the deleterious mutations that arise in high-burden systems[1–3]. For any given gene, its burden is in the first instance the resource cost of expressing the gene[4–7] and if the gene encodes a function, for example an enzyme, the impact of this can further cause a more specific role-based metabolic burden that adds to the expression burden, e.g., by consuming host cell metabolites and co-factors[8,9]. Research primarily in the model bacteria *E. coli*, has demonstrated that a lack of understanding of the burden of expressing additional genes affects our ability to predictively engineer cells[10–13].

The burden of expressing synthetic constructs is caused by competition between construct genes and host cell genes for the many different resources used in gene expression. As dozens of interconnected processes and a significant fraction of all cellular machinery are involved in gene expression, efforts to quantify burden have focused on measuring its global effects (i.e., its effect on growth rate) rather than examining availability of individual resources (e.g., polymerases, ribosomes, tRNAs., etc.)[14–18].

Previously, we developed a method to quantify the burden of synthetic constructs by measuring the competition for global expression resources when these constructs are expressed in growing *E. coli* cells that contain a capacity monitor cassette integrated into their genome[4]. The capacity monitor encodes constitutive expression of the green fluorescent protein (GFP) and the GFP production rate per cell acts as a measure of the cell's capacity for general gene expression. If the cells express a burdensome synthetic construct, their capacity for gene expression (inferred from their GFP production rate per cell) decreases due to increased competition for global expression resources.

By using GFP production rate as a proxy measurement for the availability of all resources required for general gene expression we were able to quantify the burden of different constructs and determine how choices in construct design (e.g., the promoter and RBS sequences) can give constructs with the same levels of expression but with different burdens that cause their host cells to grow at different rates[4]. We accompanied this work with a simplified mathematical model of translation focused on ribosome flow along mRNAs that captures competition for resources between synthetic constructs and a capacity monitor[19]. Simulations with this model were able to predict the outcomes of altering synthetic construct mRNA levels (via the promoter) and translation initiation efficiency (via the RBS) and also the impact of having poor codon optimization[4].

As software tools now exist to define promoter and RBS sequences of desired strengths[20–22] it thus becomes exciting to consider that the burden of synthetic constructs could be predicted from DNA sequence. With a ribosome flow model, this would only require knowledge of four parameters: the mRNA length, its abundance (i.e., promoter strength), the RBS strength and the efficiency of elongation steps taken during translation. Unfortunately, translation efficiency is currently impossible to predict from DNA sequence due to the highly complex nature of protein synthesis, which is known to be affected by nucleotide composition[23], mRNA secondary structure[24], translational pausing[25], the presence of rare codons, the use of rare amino acids[26,27], or in most cases combinations of all of the above and more. This inability to predict translation efficiency from sequence therefore emphasises a critical need for rapid ways to instead measure it.

To tackle this problem, we set-out here to develop an accessible method to quickly determine the relative cost of translation of any given protein coding sequence in *E. coli*, so that this measurement could be used as a lumped parameter representing translation efficiency in our predictive model. This would require comparing the burden imposed by constructs that have different protein coding sequences but all have the same standard promoter and RBS parts. Although this could be done in vivo using our previously described capacity monitor assay[4], we instead focused here on establishing this approach in cell-free *E. coli* lysates in order to avoid the need for growth-based experiments, which can generate hard-to-deconvolute results as burden itself slows growth and promotes mutations[1]. Instead, cell-free lysates represent a simpler, non-growing expression system that effectively captures a snapshot of the *E. coli* gene expression machinery.

Recent work has shown that cell-free lysates can greatly accelerate synthetic biology as the expression from many constructs can be characterised within hours simply by adding synthetic DNA directly to pre-prepared or purchased lysates[28–32]. Furthermore, several studies have shown that the expression of proteins using cell lysates matches in vivo expression of the same constructs in *E. coli*[28–30]. Therefore, in this work, we investigated how cell lysate experiments could be setup to replicate the competition for expression resources seen naturally within growing *E. coli*. This led to the development of a cell-free capacity assay that predicts burden in vivo and provides a lumped measure of the translation efficiency of a protein coding sequence. This measure was then used in a modified version of our ribosome flow model in order to predict the in vivo burden of further constructs, including those with different RBS strengths, different promoters and with multigene operons encoding metabolic pathways. Together these efforts offer a method where rapid in vitro screening of genes of interest in a standardised plasmid enables prediction of their expression and burden when they are implemented in different synthetic constructs in growing *E. coli* cells.

## Results

**Cell lysate conditions for prediction of in vivo burden**. Our previously described in vivo capacity monitor assay measures burden by quantifying the competition for resources in growing *E. coli* constitutively expressing GFP[4]. The measured capacity was strongly predictive of the subsequent *E. coli* growth rate (Fig. 1a). Equivalent competition for gene expression resources has also been seen in cell-free experiments when using two different plasmids in the same cell lysate mix[33]. Therefore, to measure burden in cell lysates, we constructed a capacity monitor plasmid designed for in vitro use by making a low-copy version of our existing capacity monitor cassette with constitutive *superfolder gfp* expression and a strong RBS. As with the in vivo assay, we used this as a means to characterise resource competition by measuring the capacity for expression in *E. coli* cell lysates (Fig. 1b). For this, we define the in vitro capacity as the maximum GFP production rate calculated from the GFP fluorescence of the lysate (max d$GFP$/d$t$, see Supplementary Fig. 1A).

We first measured in vitro expression in 10.5 μl of cell lysate mix, with no competing plasmid, instead using increasing concentrations of the capacity monitor plasmid itself to determine the available capacity of the lysate. This revealed that the in vitro capacity reaches a plateau at 50 nM of plasmid DNA and goes on to decrease at higher DNA concentrations (Fig. 1c). Calculating the in vitro capacity per DNA concentration (Supplementary Fig. 1B) highlights that there is spare capacity in the lysate assay below 30 nM total plasmid DNA, saturation at around 50 nM and decreased capacity per DNA above this, due to competition for expression within the pool of plasmids.

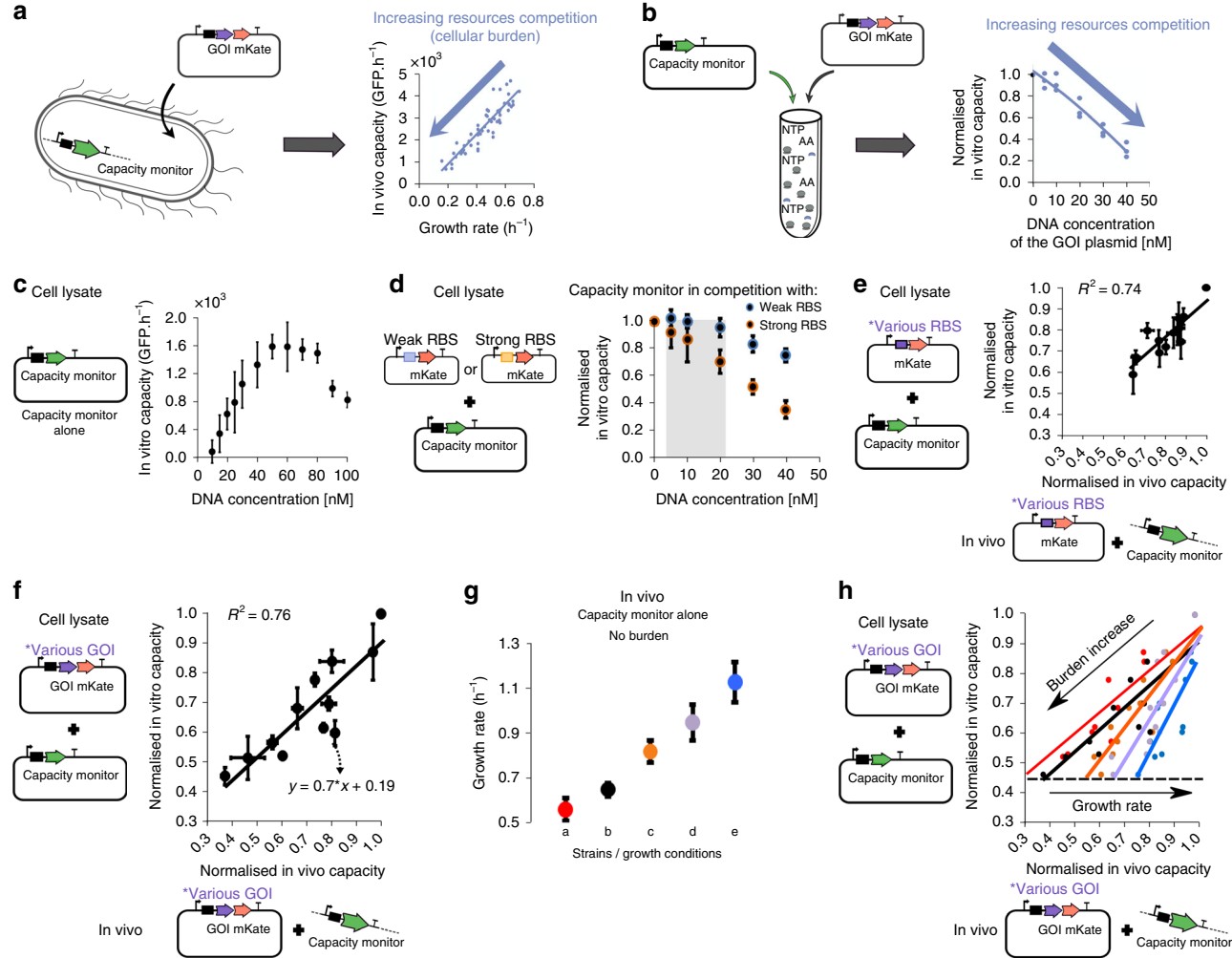

**Fig. 1** A method to measure resource competition using a capacity monitor in cell lysate. **a** Illustration of resource competition in *E. coli* between a genome-integrated GFP capacity monitor gene and a plasmid-based gene of interest (GoI) fused to *mkate*. Graph shows the correlation between the inferred expression capacity (measured as max GFP production rate per cell) and the cell growth rate. **b** Illustration of resource competition in cell lysates expressing the capacity monitor from a plasmid and the GoI from another plasmid. Graph shows the correlation between the normalized in vitro capacity (measured as max GFP production rate, Supplementary Fig. 1A) and increasing concentrations of a GoI plasmid. **c** Measured in vitro capacity with the capacity monitor plasmid added at different concentrations in cell lysate. **d** Normalized in vitro capacity measured in cell lysate containing 30 nM of the capacity monitor plasmid and different concentration plasmids bearing *mkate* with either a strong (black/orange) or a very weak (black/blue) RBS sequence. The grey area represents the concentration of plasmid where competition is for translational resources. Values are normalised to the in vitro capacity obtained with capacity monitor plasmid alone. **e** Correlation between normalized in vitro capacity measured in cell lysate and normalized in vivo capacity measured with DH10B cells. The constructs in this experiment all express *mkate* with different RBS sequences (Supplementary Table 1). **f** Correlation between normalized in vitro capacity measured in cell lysate and normalized in vivo capacity measured in DH10B using constructs with various genes of different sizes paired with RBS BCD2[64]. **g** Growth rate of strains growing in different conditions and containing only the capacity monitor. Strains/conditions: DH10B in M9 pyruvate (a, red); DH10B in M9 fructose (b, black); MG1655 in M9 fructose (c, orange); DH10B in M9 glucose (d, purple); DH10B in LB (e, blue). **h** Correlation between normalized in vitro capacity (from **f**) and normalized in vivo capacity when constructs with various genes of different sizes paired with RBS BCD2 are expressed in the strains and conditions described in **g**. Error bars show standard error of three independent repeats

In *E. coli* the main cost of gene expression is attributed to translation[6,17,18,34,35]. However, in cell lysates, while NTPs and amino acids are added in excess, polymerases, ribosomes and their associated machinery (sigma factor, tRNAs, chaperones, initiation and release factors) are added at an unknown amount and thus the relative costs of transcription or translation are not known. We therefore next set out to determine the contributions of transcription and translation to resource competition in cell lysates. We introduced two different plasmids to each compete with our capacity monitor plasmid. The first contains the *mkate* gene, paired with a constitutive promoter (BBa_J23106) and a strong RBS, and was used to measure the cost of both transcription and translation. The second plasmid has the same

genetic content but with a very weak RBS that produces no measurable mKate protein. This, therefore, imparts a transcriptional cost but negligible translational cost compared to the first plasmid. We added different concentrations of each plasmid to the cell lysate mix along with 30 nM of the capacity monitor plasmid and measured the corresponding in vitro capacity via the GFP production. Values were normalised to the in vitro capacity when the capacity monitor plasmid alone was present (i.e., normalised in vitro capacity = 1.0 in a cell lysate with only 30 nM of the capacity monitor plasmid).

A negligible decrease in in vitro capacity was observed with addition of up to 20 nM of the weak RBS plasmid, implying no competition for transcriptional resources at these concentrations

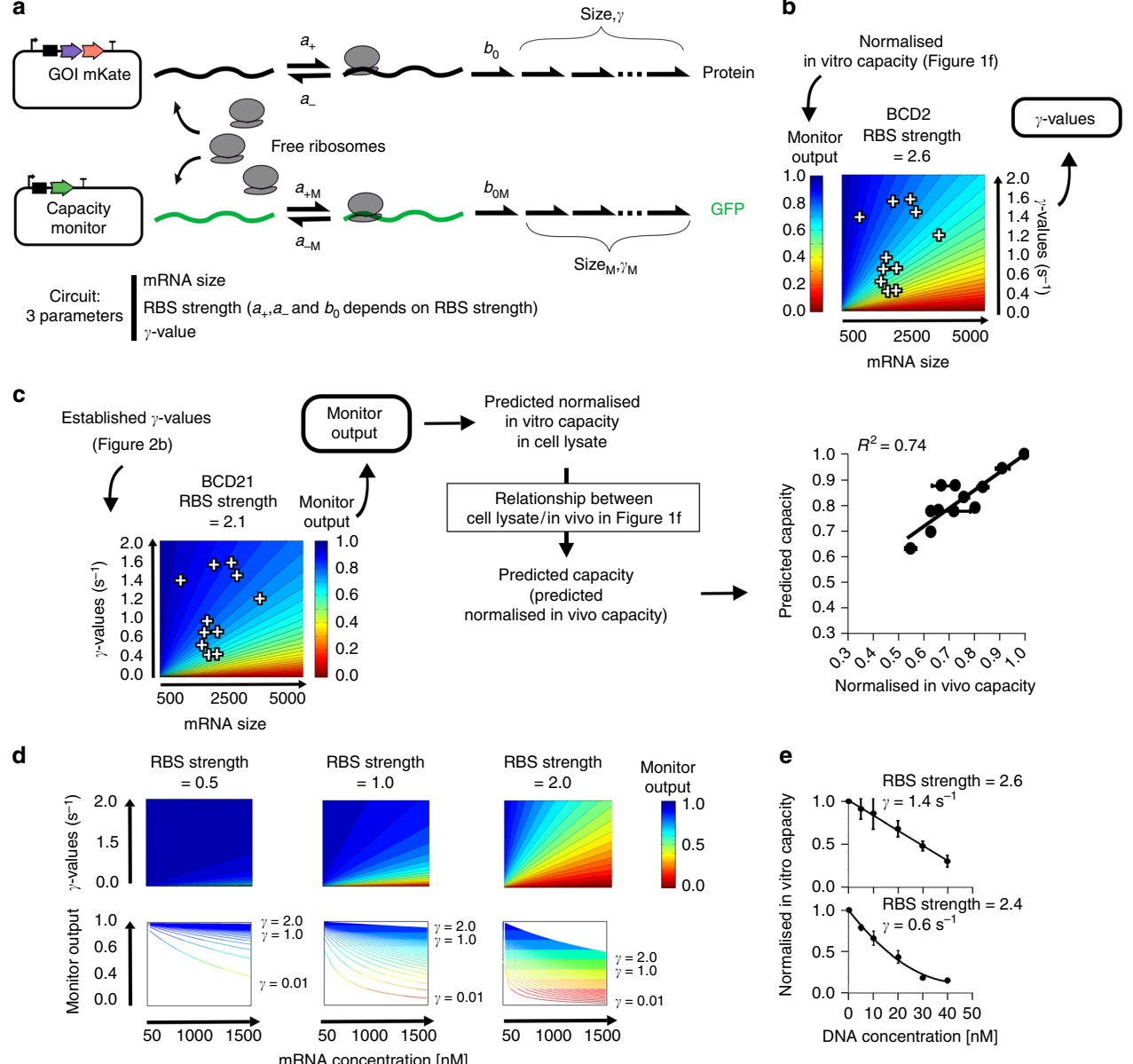

**Fig. 2** Predictive model for resource competition between a synthetic construct and capacity monitor. **a** Schematic of mathematical model of competitive translation between capacity monitor construct and a construct expressing a Gol, with a finite ribosome pool. A free ribosome binds to an unoccupied RBS at a rate $a_+$ ($a_+ = a_{+M} \times$ RBS strength) and either unbinds and returns to the free ribosome pool at a rate $a_-$ ($a_- = a_{-M}/$RBS strength), or initiates synthesis at a rate $b_0$ ($b_0 = b_{0M} \times$ RBS strength). Once synthesis has initiated, all the mechanisms of the protein synthesis are gathered in the lumped parameter $\gamma$, which represents the cost of protein synthesis. The number of protein synthesis steps depends on size (mRNA size/30—as 30 bases represents the footprint of a ribosome). Each synthesis step is considered to proceed with the same cost $\gamma$. **b** Heat maps of simulated capacity monitor expression (monitor output) when mRNA size and $\gamma$ value of a synthetic construct are varied. The heat map is used to determine the $\gamma$ value of each construct in Fig. 1f. **c** A new heat map simulated for weaker RBS is used to predict monitor output using $\gamma$ values calculated in **b**. As prediction is done for cell lysate, in vivo predictions are then deduced from the relationship between cell lysate and in vivo measurements as per Fig. 1f. **d** First row: heat maps of simulated capacity (monitor output) when mRNA concentration and $\gamma$-values of a synthetic circuit are varied. Each heat map is a construct with different RBS strength. Second row: capacity (monitor output) as a function of mRNA level. Each line corresponds to different $\gamma$-values. **e** Measurements of normalised in vitro capacity when increasing synthetic construct DNA is added to cell lysate. Top: *mkate* with a strong RBS. Bottom: *viob-mkate* with a strong RBS. The calculated $\gamma$ and RBS strengths (from Supplementary Fig. 4B) are shown for each construct. Lines show a fit to data points. Error bars show standard error of three independent repeats. Values are normalised to the capacity obtained with capacity monitor plasmid alone

(black/blue dots, Fig. 1d). In contrast, addition of the strong RBS plasmid at 20 nM gives a significant decrease in normalised in vitro capacity (black/orange dots, Fig. 1d) revealing competition for translational resources. Taken together, these results demonstrate that when our cell lysate assays are run with 20 nM of test plasmid plus 30 nM of capacity monitor plasmid,

expression is close to saturation (50 nM total DNA) and translational resources are the major limitation, as seen in vivo.

**Correlation of protein cost between cell lysate and in vivo measurements.** Using the conditions determined above, we next

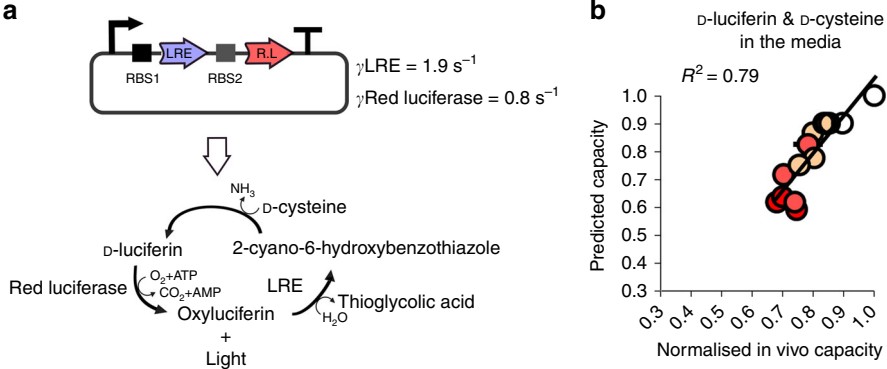

**Fig. 3** Predicting the burden of operon designs for the Luciferase biosynthesis pathway. **a** Diagram of the luciferase pathway and the $\gamma$-values for the two enzyme-encoding sequences as measured by the cell lysate capacity assay (see Supplementary Fig. 5). The operon is designed with randomised RBS sequences and promoter BBa_J23100 (strong). **b** Model-predicted burden of each operon design compared to the measured capacity of E. coli expressing the operons. D-Luciferin and D-cysteine are added to the media to enable luminescence measurements and these are represented by the red intensity in each circle. Error bars show standard error of three independent repeats. Values are normalised to the capacity obtained with capacity monitor plasmid alone

measured the burden of a collection of plasmids expressing mKate at different levels in our cell lysate assay. We constructed a library of plasmids (see Supplementary Table 1) with *mkate* under control of the same promoter but with different RBS sequences in order to alter translation initiation efficiency. These plasmids were first assessed by our previous in vivo capacity monitor approach (GFP production rate per cell) and shown to cause more burden as increased mKate is expressed (Supplementary Fig. 2A) with the decreased in vivo capacity correlating strongly with decreased E. coli growth rates ($R^2 = 0.88$, Supplementary Fig. 2B). We then measured these plasmids in our cell lysate assay to obtain their in vitro capacity measurements and compared these to the equivalent in vivo data. This revealed a well-correlated linear relationship between the in vitro and in vivo capacity measurements ($R^2 = 0.74$, Fig. 1e) that suggests that the resource competition seen in vitro is broadly predictive of that seen in vivo for most cases.

Having demonstrated predictability in cases where translation initiation rate is altered, we next looked to see if cell lysate assays can predict the burden of producing different proteins. First, we constructed a standard entry vector to enable the protein coding sequence of any gene of interest (GoI) to be rapidly cloned by Golden Gate DNA assembly into a standard format for use in the cell lysate assay (see Supplementary Table 2, Supplementary Fig. 1C). This design leads to the GoI protein coding sequence being constitutively expressed as a fusion protein with C-terminal mKate (in order to allow expression to be verified). We constructed 3 different entry vectors, each with different RBS sequences in order to give a choice of expression levels. We selected the well-characterised B0034 RBS along with two Bicistronic Design (BCD2 and BCD21) sequences that ensure context-free, defined levels of translation initiation[36].

The protein coding sequences of 7 arbitrarily chosen genes of different lengths, functions and amino acid composition and 3 truncated versions of *viob* of different lengths were all cloned into the same entry vector with the BCD2 (Supplementary Table 2). When assayed both in cell lysate and then in E. coli, a wide range of burden was observed for this collection. The capacity monitor measurements from both cell lysate and in vivo experiments once again showed a good linear fit, with a $R^2 = 0.76$ (Fig. 1f), demonstrating that the resource limitations in cell lysates for translating different proteins (and transcribing their different mRNAs) are similar to the resource limitations seen in E. coli. This linear relationship between the data offers a route to using cell lysate measurements to

predict the burden of expressing any protein of interest in vivo in growing cells.

To complete the verification of our approach, different strains and growth conditions were next chosen to investigate whether our standard in vitro lysate assay can predict burden in different in vivo environments. As translation and transcription machinery concentrations are known to be condition-dependent in vivo, we anticipated that the competition for resources would vary when experiments were done with E. coli growing at different rates[37–40]. To achieve this, we first determined different growth media and strain combinations that yield a range of growth rates of unburdened cells containing the in vivo capacity monitor (Fig. 1g). The 10 GoI-expressing constructs used above (Fig. 1f) were then re-assayed by the in vivo capacity monitor assay in these different conditions and compared to previously taken in vitro capacity measurements (Supplementary Fig. 3). Once again, well-correlated linear fits were seen in all conditions, but notably the slopes of these fits are different. When superimposed together, a clear trend can be seen where the slope of the relationship between in vitro and in vivo measurements increases with increasing growth rate (Fig. 1h). The burden in vivo thus appears to be less important in richer media and faster growing cells.

**Predicting in vivo burden from cell lysate measurements**. Having established a standard cell lysate assay, we next integrated this into existing efforts to mathematically model the relationship between gene expression and burden. We used a simplified version of the model of translation developed by Algar et al., that was previously experimentally tested[4,19] (Fig. 2a). This model (outlined in Supplementary Note 1) describes the three main steps of translation: initial binding of ribosomes, protein synthesis and ribosome release. The total number of ribosomes—a global parameter meant to account for all the translation resources being competed for—is fixed in this model and is also expected to be fixed in cell lysate experiments. The binding ($a_+$), unbinding ($a_-$) and synthesis initiation ($b_0$) rates of ribosomes on an mRNA all depend on the RBS strength (see Supplementary Note 1 and Supplementary Fig. 10). The amount of resources needed to produce a protein is captured by a lumped parameter $\gamma$, which represents the translational cost of synthesising the protein. The value of $\gamma$ will vary for each GoI depending on the protein being made and how efficiently it is translated.

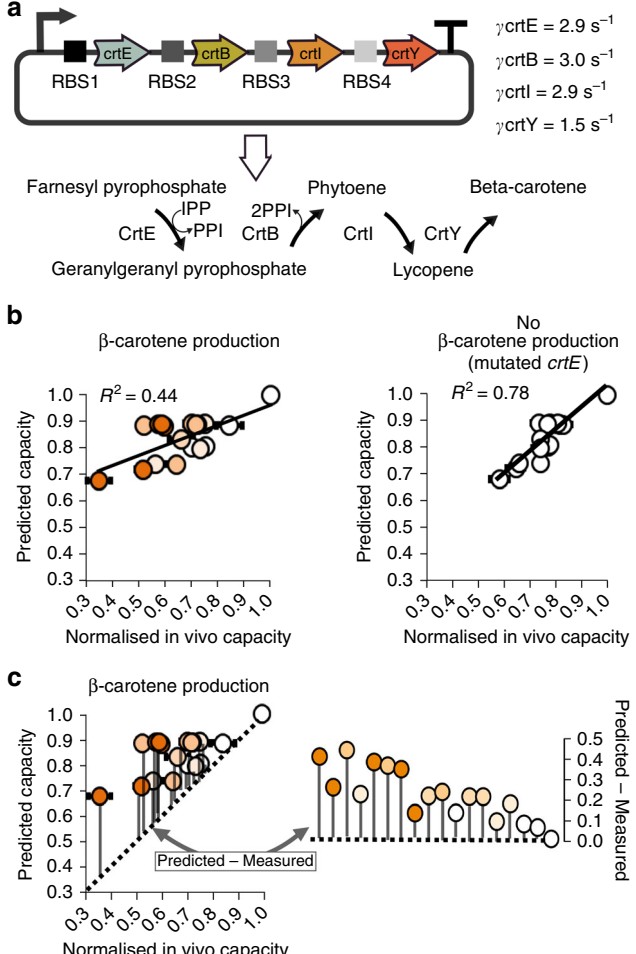

**Fig. 4** Predicting the burden of operon designs for the beta-carotene biosynthesis pathway. **a** Diagram of the beta-carotene pathway and the $\gamma$-values for the four enzyme-encoding sequences as measured by the cell lysate capacity assay (see Supplementary Fig. 7). The operon is designed with partially randomised RBS sequences and one of three promoters: BBa_J23113 (weak), BBa_J23106 (medium), or BBa_J23100 (strong). **b** Model-predicted burden of each operon design compared to the measured in vivo capacity of *E. coli* expressing the operons with or without an inactivating mutation in the *crtE* gene (prediction method described in Supplementary Fig. 8). The orange intensity in each circle represents the measured beta-carotene level for each strain (see Supplementary Fig. 9). Error bars show standard error of three independent repeats. **c** Model-predicted burden of each operon design compared to the measured in vivo capacity of *E. coli* expressing the active pathway (same data as **b**, non-mutated pathway on left). The diagonal dot line represents equality between predicted and measured normalised in vivo capacity. Grey bars indicate the difference between the predicted and measured normalised in vivo capacity of the 17 operons. Right plot compares the relative differences between predicted and measured normalised capacity for the 17 operons and the strain-only control. Operons are ranked from low to high-in vivo capacity values. Values are normalised to the capacity obtained with capacity monitor plasmid alone

Our lysate assay measurements showed that when mRNA levels and RBS strengths are kept the same, capacity can still vary considerably due to changes in the protein coding sequence of the GoI (Fig. 1f). In the model used here, these changes only alter two parameters, the mRNA size and $\gamma$. Given that the mRNA size is known for any GoI, this means that $\gamma$ can be readily estimated from in vitro capacity measurements. To demonstrate this, we

modelled the competitive expression of the 10 GoI test constructs from Fig. 1f. As these constructs all have a standardised strong RBS (BCD2), the RBS strength in the model was set to a value of 2.6 (see Supplementary Fig. 4). Other parameters in the model were then set so that the simulated monitor output from the model matched the normalised in vitro capacities we observed in the cell lysate data. The $\gamma$-value for each GoI could then be determined from these parameter settings once the known mRNA size is taken into account (Fig. 2b). In other words, via this method $\gamma$ is immediately deduced from cell lysate measurements when mRNA size and RBS strength are known.

To test the predictive power of our approach, we then used the model-inferred $\gamma$-value for each of these 10 GoIs in simulations of equivalents constructs that have BCD21, a weaker RBS that we experimentally measured to correspond to RBS strength 2.1 (Supplementary Fig. 4). Using this simulation (Fig. 2c), we first predicted the cell lysate normalised in vitro capacity for these new constructs, and then using the known linear relationship between lysate and in vivo measurements ($y = 0.7x + 0.19$ from Fig. 1f), we extended this to predict the in vivo performance (Fig. 2c). We then built and measured the burden of this BCD21 library in *E. coli* and compared predicted in vivo performance with the in vivo capacity data, again seeing a good correlation ($R^2 = 0.74$, Fig. 2c). Thus, with only the cell lysate data and knowledge of the mRNA length and RBS strength, we are able to predict the burden of different genes of interest expressed at different levels in *E. coli*.

Further investigation of our model revealed that modification of our parameters (RBS strength, $\gamma$-values) lead to different capacity monitor outputs profiles. In Fig. 2d, the mRNA concentration varies to simulate an increase in the copy number of a GoI and what impact this has on the monitor output. The mRNA amount and the monitor output (capacity) exhibit a linear relationship at very low-RBS strength (e.g., 0.5) and $\gamma$-value higher than $0.02\,\mathrm{s}^{-1}$ (note that the $\gamma$-value for the monitor *gfp* gene is $1\,\mathrm{s}^{-1}$). In this context the impact of the translation of several genes is *additive* (i.e., the decrease in monitor output will be the sum of the decreases in monitor output values for each gene measured individually).

However, at high-RBS strengths a decrease of the $\gamma$-value leads to a faster-than-linear decrease in the monitor output as mRNA amount increases (e.g., RBS strength 2 in Fig. 2d). Using cell lysate measurements, we experimentally demonstrated this effect by comparing the normalised in vitro capacity from the monitor construct when it competes against expression of a gene with a high $\gamma$-value (*mkate*, $\gamma = 1.4\,\mathrm{s}^{-1}$) and with a strong RBS (RBS strength 2.6) vs. competing against expression of a gene with a low-$\gamma$-value (*viob-mkate*, $\gamma = 0.6\,\mathrm{s}^{-1}$) and a strong RBS (RBS strength 2.4). To mimic increased mRNA levels, we simply added more DNA to the cell lysate assay for these two plasmid constructs. As predicted by our model, we saw a linear relationship for the high-$\gamma$-GoI (Fig. 2e, upper graph), and a nonlinear relationship for the low-$\gamma$-GoI (Fig. 2e, lower graph).

**Predicting the burden of a two-gene operon**. To demonstrate that the cell lysate plus model approach can be used to predict the in vivo burden for a multigene system, we characterised and simulated a two-gene luciferase operon used for generating bioluminescence in *E. coli*. This operon consists of two genes from the firefly *Luciola cruciate* (Fig. 3a). Using our cell lysate assay we first measured the in vitro capacity when each gene is expressed individually and used this to determine their $\gamma$-values (Fig. 3a). This was done with both the standard GoI-testing vector (with BCD2 RBS) and further confirmed with a second vector with B0034 RBS (Supplementary Fig. 5). We then took an existing plasmid construct where these two genes are in an operon

expressed from a constitutive promoter, and randomly modified the two RBS sequences in this to obtain a collection of 14 operons where expression of the two enzymes is at a variety of levels. We then estimated RBS strengths via the RBS Calculator[20,21] using this to guide the RBS parameterisation in our model (see Supplementary Fig. 6). As the promoter used in the construct has been previously characterised, an estimate for mRNA concentration is also known for all members of the library (see Methods section, Model simulations).

Expression of all 14 library members was then simulated in the capacity monitor model using the cell lysate-determined $\gamma$-values, the known mRNA lengths, and the sequence-estimated mRNA levels and RBS strengths. We then compared the model-predicted impact on capacity with subsequent capacity monitor assay measurements taken in E. coli expressing all 14 bioluminescence constructs (Fig. 3b). We observed a strong correlation ($R^2 = 0.79$), this time demonstrating that a multigene system can be predicted from sequence information and standard cell lysate measurements.

**Predicting the burden of metabolic pathway operon.** To further verify our approach, we attempted to predict the in vivo burden of a library of designs of a more complex operon encoding the four gene metabolic pathway for the biosynthesis of beta-carotene, a metabolite of interest for medicine and used industrially in nutritional supplements, cosmetics and animal feed[41,42]. We took genes from Erwinia uredovora[43,44] codon optimised these for E. coli and characterised the individual burden of each gene's expression in our standard cell lysate. This gave us the $\gamma$-value for each enzyme (Fig. 4a, Supplementary Fig. 7). We then used Golden Gate DNA assembly[45] to construct a collection of 17 operons expressing all four enzymes at a variety of levels with one of three different constitutive promoters and with partially randomised RBS sequences for each enzyme-encoding GoI (Fig. 4a, Supplementary Fig. 8A). For model-based predictions, RBS strengths were estimated from sequence information as before and mRNA levels were estimated from the known relative strengths of the promoters used. The 17 operons were then assessed in vivo by E. coli capacity monitor assay and also assessed for beta-carotene production by colour imaging of cell pellets (Supplementary Fig. 9A).

For the first round of predictions, we made the initial assumption that the total burden of expressing the operons in vivo would entirely be due to the cost of expressing the genes, i.e., we assumed no significant impact on metabolism via conversion of host metabolites into beta-carotene. However, when we compared the predicted effect on capacity from our model with subsequent measurements taken in E. coli (Fig. 4b) we only saw a weak correlation ($R^2 = 0.44$) while also observing a wide diversity in beta-carotene production from the different designs (Supplementary Fig. 9, Supplementary Note 2). Thus, it became evident that the in vivo burden of each operon must also relate to the metabolic cost of running the pathway in the host cell.

Expression of the pathway enzymes depletes the cell of key metabolites, which presumably affects cell growth, such as the precursors farnesyl pyrophosphate (FPP) and isopentenyl pyrophosphate (IPP) both involved in terpenoid backbone synthesis[46]. To investigate this, we targeted a mutation to the active site of the first enzyme of the pathway, CrtE, in order to inactivate it and effectively cease metabolic conversion for the whole pathway. The mutation was designed to have no effect on the expression of the enzymes and so gene expression burden was still seen when the 17 operons, each with this mutation, were re-characterised in vivo for their effect on E. coli capacity. With no observable

beta-carotene production, our in vivo data now showed a much-closer match to our initial predictions for the burden of each design ($R^2 = 0.78$, Fig. 4b). Our cell lysate assay-based model is thus able to predict the impact on the host of expressing multiple genes but gives the most accurate predictions when burden is only the result of competition for gene expression resources. Interestingly, this means that the burden caused by the roles of enzymes can also be estimated by subtracting the predicted in vivo capacity from the subsequent capacity of cells measured in vivo. The resulting difference calculated from this gives a value of burden that is not predicted to be caused from gene expression alone (Fig. 4c, Supplementary Fig. 9, discussed in Supplementary Note 2). Excitingly, this finding means that our approach offers potential further use for separating expression burden from metabolic burden.

## Discussion

This work demonstrates that a standard cell lysate-based assay can be used to quantify the burden of expressing a protein coding sequence and provides an otherwise missing parameter for predicting the burden synthetic gene expression places on E. coli. We demonstrated here that competition for translational resources in cell lysates serves as a good predictor for in vivo behaviour in E. coli. Furthermore, we provide a standard entry vector to enable quick, standardised characterisation of a GoI with cell lysates. When combined with promoters of known strengths and estimates of RBS strength from the RBS Calculator[20,21] the lysate measurements and model can predict the burden of different single-gene constructs and of multigene contructs. Given the broad biotechnological importance of engineered protein expression from E. coli, we anticipate that this approach will have wide interest, enabling those focused on genetically engineering cells to design expression constructs that take into account how these will impact on host cell growth.

In particular, measurement using cell lysates offers several advantages. While all constructs assessed in this study were plasmids cloned via E. coli, others have demonstrated that reliable cell lysate measurements are possible from directly synthesised DNA or PCR amplicons, avoiding the need for any in vivo steps[28]. Cell lysate assays are also quick and more readily miniaturized, having been shown to work in microfluidic systems and in 384-well microplate format[29,32,47]. However, since they are only a snapshot of expression resources of growing E. coli they cannot show the same dynamic behaviours of growing cells that can regulate gene expression in the face of burden. Yet for measurement, this lack of dynamic adaptation is actually of benefit, enabling resource competition to be determined in a standard assay without the confounding factors of growth rate changes or changes in gene regulation coming into play. Parallel work by our group examining how E. coli adapts to burden has recently shown that host stress responses that change global gene expression are activated within minutes of synthetic gene expression induction[48].

While the cell lysate measurements showed good predictive power ($R^2 > 0.7$) for the equivalent in vivo behaviour in our work, the correlations were not perfect. The variance observed when comparing cell lysate and in vivo data (Fig. 1e) will likely stem from two things; the variance of experimental measurement, and the fact that some protein coding sequences will be affected in vivo by rapid, dynamic processes (e.g., stringent response) that may not operate in the same way in vitro. A further interesting observation was that when lysate is saturated with DNA (>50 nM plasmid) we see an unexplained decrease in expression per plasmid (Fig. 1c). While further work would be required to understand this phenomenon, one possibility is that saturating

transcription can inhibit translation (or vice versa) by consuming a limited resource required by both (e.g., GTP).

Notably, our model focuses on translation and so does not capture every effect related to burden, such as cases where transcriptional resource competition comes into play. The model also cannot predict the effects of burden in stationary phase or if the constructs being simulated contain sequences that specifically change behaviour in response to burden or growth changes (e.g., stress-response promoters). Yet despite this, our approach still offers good predictability ($R^2 > 0.7$), even for libraries of multi-gene constructs. While stronger correlations would be more desirable it is notable that these values are already on par with the best accuracy achieved so far with RBS Calculator models[20,21,49]. Indeed, RBS strength prediction is likely to be the bottleneck for accuracy in our approach given the importance this sequence plays in determining gene expression levels and translational burden.

Another interesting outcome from this work is the possibility that our approach can separate expression burden from metabolic burden, something that cannot easily be done in vivo due to the combined effects that all types of burden have on host cell growth rate. Our characterisation of beta-carotene pathway operons demonstrates that these two types of burden are jointly responsible for decreased growth rates of hosts expressing heterologous genes to produce metabolites. Most methods for metabolic pathway optimisation seek to produce the most product while doing so with the minimal cost of expression of the enzymes[50–52]. Therefore, quantifying the individual contributions to burden of both gene expression and pathway productivity offers an additional tool for designing the most productive pathways. Further research into this could aid a future study to optimise productivity and growth trade-offs for a metabolic pathway known to require the expression of difficult-to-translate enzymes, such as polyketide synthases and non-ribosomal peptide synthetases.

Ideally, our lysate measurement approach will only need to be a temporary method that will one day be superseded by an ability to predict translation efficiency directly from sequence. Several major efforts have now assessed the effects of combinatorial sequence changes on the expression of GFP in E. coli in order to determine which factors have the greatest effect on expression efficiency. Initially this research concentrated on features at the 5′ end of GFP-encoding mRNA[53,54], but the most recent study of over 240,000 designs extends analysis of sequence space into the protein coding sequence and now combines growth assays and ribosome profiling data to further our understanding of how translation efficiency and burden are related[55]. While providing considerable new information, this latest research further underlines that multiple different factors and process combine and even act upon each other to determine translation efficiency. Indeed, this study concludes that it is currently impossible to predict the cost of protein production from the DNA or protein amino acid sequence alone[55].

Future iterations of our cell lysate approach could also aid in further understanding the mechanisms of expression burden and resource competition at a molecular level. Using defined in vitro expression systems such as PURE Express (NEB), which contain known quantities of purified components, such as polymerases and ribosomes[56], would allow full control of the make-up of cell lysates and provide a route towards determining the main components that are required for efficient gene expression. This could help investigate which factors are limiting for different genes. For example, charged tRNAs may be limiting for genes with rare codons, while chaperones may be limiting for genes requiring complex folding. Such an approach would likely reveal hidden mechanisms and constraints in gene expression, highlighting basic components to increase in cells when needing to efficiently

overexpress certain genes, while also providing a more complete list of the components needed for the construction of minimal cells.

## Methods

**Strains and growth media**. Plasmids were transformed using standard procedures[57] in chemically competent E. coli DH10B-GFP or MG1655-GFP. DH10B-GFP is the DH10B strain with genome integrated capacity monitor that consists of a strong constitutive promoter (BBa_J23100), strong synthetic RBS (`tacta-gagaaatcaaattaaggaggtaagata`), a codon-optimized superfolder GFP[58] coding sequence, and a synthetic unnatural bidirectional terminator, integrated in the λ loci of E. coli genomes[4] (the same design is used for MG1655-GFP). Bacterial growth was performed at 37 °C in minimal media M9 supplemented with 0.5% fructose (or 0.5% glucose or pyruvate or LB media, see Fig. 1g and Supplementary Fig. 3) and chloramphenicol (35 µg/ml).

**Construction of the GoI-mKate library**. The chloramphenicol-selectable, high-copy plasmid pSB1C3 (BioBricks Foundation) was used as a backbone to construct the standard entry vector for GoI insertion (Supplementary Fig. 1C). To construct this vector we first PCR amplified pSB1C3 (Forward: `taagccagccccccga-cacccg`, Reverse: `tgaaccacagagtgattaat`) and lacZ under control of pLac promoter flanked by BsaI restriction sites (Forward: `gcagctggcacgacaggttt`, Reverse: `ttatgcggcatcagagcaga`). Second, the linker-mKate sequence was codon optimised and ordered for synthesis by GeneArt with the RBS sequences BCD2 and B0034. The different parts were then assembled and cloned using the Gibson Assembly method[59] to obtain the standard entry vector described in Supplementary Fig. 1C.

The selected GoI (Supplementary Table 2) were all obtained from BioBrick format DNA from the iGEM Parts Registry. This was used as template and PCR amplified to be flanked by appropriate BsaI restriction sites (`ggtctcannnn`). Golden Gate assemblies were setup by pipetting 40 fmol of backbone and insert, 0.5 µl of BsaI (NEB UK), 0.5 µl of T7 DNA ligase (NEB UK), 1 µl T4 buffer (NEB UK) and completed with water for final volume of 10 µl. Then the mix was put in a thermocycler for 30 following cycles: 42 °C for 2 min/6 °C for 5 min/55 °C for 1 h/80 °C for 10 min.

**Construction of the luciferase operon library**. The luciferase operons were constructed by modifying the arabinose-inducible luciferase operon of Luciola cruciatae (http://parts.igem.org/Part:BBa_K325219). First, the BBa_J23100 promoter was inserted upstream of the luciferase to replace of pBAD by whole-plasmid inverse PCR amplification with primers:

Forward: `cctaggtacagtgctagctactagagttaaggaggtaa`, Reverse: `actgagctagccgtcaactctagaagcggccgcgaat`. The resulting PCR amplicon was ligated using T4 DNA ligase and transformed in DH10B cells. Next, 2 rounds of PCR amplification were used to obtain a collection of constitutively expressed operons with a wide variety of RBS sequences. The first round used the following primers: Forward: `rrrrrrnnnnnatggccccgaccgtggaaca`, Reverse: `taactctagtactctagaag`. The resulting PCR fragments were ligated using T4 DNA ligase and transformed in DH10B. The colonies obtained were pooled and grown in LB media overnight at 37 °C. Plasmids were then extracted using Miniprep kit (QIAGEN Plasmid Minipreps Kit) and used as template for the second whole-plasmid inverse PCR amplification using primers: Forward: `rrrrrrnnnnnatggagaacatggagaacga`, Reverse: `taaggatccttattacagct`. The resulting PCR fragments were ligated using T4 DNA ligase and transformed in DH10B to obtain the final library.

**Construction of the beta-carotene operon library**. The beta-carotene operons were build using MoClo toolkit[45]. The Level 0 library is composed of constitutive promoters of the Anderson collection (BBa_J23114, BBa_J23113, BBa_J23100, BBa_J23106 and BBa_J23115), a random collection of RBS sequences, the 4 enzymes of the beta-carotene pathway and the terminator T1[45]. The level 1 constructs were designed to put each enzyme under the control of a random RBS and to place the genes of the beta-carotene operons in the following order at the final level: crtE, crtB, crtI and crtY. Cloning was done using Golden Gate assembly methods with transformation into DH10B-GFP.

Two constructs (B3-1 str and B10-2 str, both containing promoter BBa_J23100) were obtained by PCR amplification designed to introduce mutations in the weak promoter sequence BBa_J23113. This was done by using whole-plasmid inverse PCR amplification with the primer pair Forward: `tacggctagctcagtcctaggtatagtgctagcgcaagggcccaag` and Reverse: `ttcacagagtggcctcgtga` using previously obtained constructs (B3-1 and B10-2) as templates. The resulting PCR fragments were ligated using T4 DNA ligase and transformed in DH10B-GFP.

All the mutated crtE constructs were obtained by whole-plasmid inverse PCR amplification (Forward: `gccgctatgccctgcatggacg`, Reverse: `cgcggcttcgctgatccctt`) followed by T4 DNA ligation in order to introduce mutations in crtE leading to an inactivated CrtE enzyme. The mutation of crtE (from `gacgat` to `gccgct`, amino acids DD to AA) was chosen in order to

modified the active site of *crtE* deduced by sequence homology of *crtE* sequences from *Erwinia uredovora*[60], *Erwinia herbicola*[60], *Rhodobacter capsulatus*[60], *Arabidopsis thaliana*[61,62] and *Euglena gracilis*[62].

**Cell lysate mix preparation and reactions.** The cell lysate preparation is based on the protocol of Sun et al.[32]. Briefly, the protocol of Sun et al.[32] is a 5 day protocol in three phases: harvest cells (colonies grow on plate over night at 37 °C, 50 ml preculture at 37 °C during 8 h, 4 liters of cultures at 37 °C until $OD_{600}$ = 1.5-2.0), extract preparation (multiple pellet washing followed by beads-beating to obtain an extract and dialysis) and cell-free reaction optimisation (optimisation by varying the Mg-glutamate, K-glutamate and PEG-8000 concentrations). The protocol was modified by using sonication[63] instead of use of a bead beater to obtain DH5 alpha cell extracts. After washing the cells as following the Sun et al.[32] protocol (Day 3 step 18) with S30A buffer (14 mM Mg-glutamate, 60 mM K-glutamate, 50 mM Tris, 2 mM DTT, pH 7.7), the cells were centrifuged 2000×g for 8 min at 4 °C. The pellet was re-suspended in S30A (pellet mass (g) × 0.9 ml). The solution was split in 1 ml aliquots in 1.5 ml Eppendorf tubes. Eppendorf tubes were placed in a cold block and sonicated using a Vibra-Cell™ Ultrasonic Liquid Processors VCX 130 using the followings procedure:

40 s ON—1 min OFF—40 s On—1 min OFF—40 s ON. Output frequency 20 kHz, amplitude 50%.

The remaining protocol followed the procedure of the Sun et al.[32] protocol for day 3, step 37. mRNA and protein synthesis are performed by the molecular machineries present in the extract, with no addition of external enzymes. The amino acid solution and energy solution mixes are kept as in the Sun et al.[32] protocol and are added to the cell extract. Reactions take place in 10.5 μL volumes at 29 °C in 384-well plate. The final cell lysate contains 3 mM Mg-glutamate, 8 mM K-glutamate, 1.5 mM of each amino acid (except leucine), 1.25 mM leucine, 50 mM HEPES, 1.5 mM ATP and GTP, 0.9 mM CTP and UTP, 0.2 mg/mL tRNA, 0.26 mM CoA, 0.33 mM NAD, 0.75 mM cAMP, 0.068 mM folinic acid, 1 mM spermidine, 30 mM 3-PGA, 2% PEG-8000. Note that there is likely to be some genomic DNA present in this final mix, even though lysate production contains a step to digest remaining nucleic acids with endogenous exonucleases. However, this amount will be constant between samples.

**Capacity monitor assay in vivo and data analysis.** For in vivo capacity measurements in DH10B-GFP, cells were grown at 37 °C overnight with aeration in a shaking incubator in 5 ml of defined supplemented fructose M9 media with chloramphenicol (35 μg/ml). In the morning, 20 μl of each sample was diluted into 1 ml of fresh medium and grown at 37 °C with shaking for another hour. We then transferred 200 μL into a 96-well plate (Costar), placed samples in a Synergy HT Microplate Reader (BioTek) and incubated at 37 °C with orbital shaking at medium setting, performing measurements of GFP (excitation (ex.), 485 nm; emission (em.), 528 nm), RFP (ex., 590 nm; em., 645 nm), OD (600 nm) and OD (700 nm) every 10 min.

Growth were calculated using $OD_{700}$ with:
Growth rate at t2 = [ln(OD(t3)) − ln(OD(t1))]/(t3 − t1), with t2 = time of the mid exponential phase, t3 = t2 + 0.5 h and t1 = t2–0.5 h.
Protein production rates per hour were calculated with:
GFP production rate at
t2 = [(total GFP(t3) − total GFP(t1))/(t3 − t1)]/OD(t2), and RFP production rate at t2 = [(total RFP(t3) − total RFP(t1))/(t3 − t1)]/OD(t2).
The normalised in vivo capacity is the GFP production rate measured in strains with the DH10B-GFP containing the test construct plasmid, divided by the GFP production rate measured in DH10B-GFP without any test plasmid.

**Resource competition assay in cell lysate and data analysis.** For resources competition in cell lysate, reactions took place in 10.5 μL volumes at 29 °C in 384-well plate (Nunc™ 384-Well). Each reaction is a mix of 7.88 μL cell lysate (cell extract + amino acid + energy solution + Mg-glutamate buffer + K-glutamate buffer + PEG, see ref. [32]), plasmid DNA and complete with sterile water to get 10 μL. Plates were centrifuged 5 min, 563×g at 4 °C (Eppendorf, centrifuge 5810R). Samples were placed in a Synergy HT Microplate Reader (BioTek) and incubated them at 37 °C with orbital shaking at low setting, performing measurements of GFP (excitation (ex.), 485 nm; emission (em.), 528 nm), RFP (ex., 590 nm; em., 645 nm) every 5 min.

Protein expression rates per hour were calculated with:
GFP production rate at t2 = (total GFP(t3) − total GFP(t1))/(t3 − t1), and RFP production rate at t2 = (total RFP(t3) − total RFP(t1))/(t3 − t1).
The maximal expression rate value was selected as described in Fig. 1c. The normalised in vitro capacity is the in vitro capacity rate measured in a cell lysate mix containing the capacity monitor plasmid and the test construct plasmid divided by the in vitro capacity measured in a cell lysate mix with only the capacity monitor plasmid.

**Bioluminescence measurements.** DH10B-GFP with the different operons, were grown at 37 °C overnight with aeration in a shaking incubator in 5 ml of defined supplemented fructose M9 media with chloramphenicol (35 μg/ml). In the morning, 20 μl of each sample was diluted into 1 ml of fresh medium and grown at

37 °C with shaking for another hour. We then transferred 200 μL of this into a 96-well plate (Costar), placed samples in a Synergy HT Microplate Reader (BioTek) and incubated at 37 °C with orbital shaking at medium setting, performing measurements of Bioluminescence (emission (em) 645 nm) OD (600 nm) and OD (700 nm) every 10 min.

**Beta-carotene measurements.** *E. coli* were incubated with aeration in a shaking incubator in 5 ml of minimum media M9 supplemented with 0.5% fructose at 37 °C during 24 h. Cells were collected using centrifugation at 2256×g for 10 min (Eppendorf, centrifuge 5810R). Pellet was re-suspended in 300 μL acetone, homogenised by vortexing and incubated at 55 °C for 15 min. Supernatant was collected after 1 min centrifugation at 18407×g (Eppendorf, centrifuge 5424). A volume of 100 μL of water was added to 100 μL of samples and OD (450 nm) was measured in a Synergy HT Microplate Reader (BioTek).

**Model simulations.** We used the competitive model of translation developed by Algar et al.[19]. Simulations were done using key parameters obtained from Bionumbers.org and calculated by previous work[37] along parameters determined from the relative strengths of the RBS sequences used, as determined from our experimental characterization. The parameters used for the capacity monitor construct in the model have been previously described[19] and are the same in all our simulations (ribosome binding rate, $a_{+M}$ = 0.0001 rib$^{-1}$ RBS$^{-1}$ s$^{-1}$; ribosome unbinding rate, $a_{-M}$ = 200 rib-RBS$^{-1}$ s$^{-1}$; synthesis initiation rate, $b_{0M}$ = 1 s$^{-1}$; mRNA concentration = 900 nM, size$_M$ = 720 bp; and synthesis rate, $\gamma_M$ = 1 s$^{-1}$). As promoters BBa_J23100, BBa_J23106 and BBa_J23113 are known to be strong, medium and weak promoters, we assumed each copy of the promoter would produce about 30, 10 and 3 copies of mRNA per DNA molecules, respectively (estimated from Bionumbers I.D. 107667). We added 20 nM of each plasmid in our mix leading to 600, 200 and 40 nM of mRNA in the cell lysate mix. Parameters for simulation of construct expression and capacity monitor expression were as follows: total available ribosomes, 2500 nM (25,000 ribosomes per cell[37] extracted from 1.6 × 10$^{12}$ cells, stocked in a final cell extract volume of 9600 μl and diluted three time in the final cell lysate mix); capacity monitor length, 24 (720 bp/30); capacity monitor mRNAs, 900 nM (30 nM × 30 mRNA per DNA as BBa_J23100 is a strong promoter); capacity monitor $b_{0M}$ = 1 s$^{-1}$; capacity monitor $\gamma$ = 1 s$^{-1}$. The RBS strength of each construct is a relative value compare to the monitor RBS strength (RBS strength of monitor = 1). For each construct the binding ($a_+$), unbinding ($a_-$) and elongation initiation ($b_0$) rates are function of the RBS strength and of $a_{+M}$, $a_{-M}$, and $b_{0M}$, respectively. More specifically, in our model we define $a_+$ = $a_{+M}$ × RBS strength, $a_-$ = $a_{-M}$/RBS strength, and $b_0$ = $b_{0M}$ × RBS strength[19]. For the sake of clarity $b_0$ and $\gamma$ units are noted in s$^{-1}$ in the manuscript but are more precisely in (10 codons) s$^{-1}$. The $b_0$ and $\gamma$ units are (10 codons) s$^{-1}$ because 10 codons is approximately the footprint of each ribosome on an mRNA and thus for our model it better represents how many ribosomes can be queued on a transcript.

**List of primers.** pSB1C3_fw

```
taagccagccccgacacccg
```
pSB1C3_rv
```
tgaaccacagagtgattaat
```
*lacZ*_fw
```
gcagctggcacgacaggttt
```
*lacZ*_rv
```
ttatgcggcatcagagcaga
```
J231008_luci_fw
```
cctaggtacagtgctagctactagagttaaggaggtaa
```
J231008_luci_rv
```
actgagctagccgtcaactctagaagcggccgcgaat
```
RandRBS1_luci_fw
```
rrrrrrrnnnnnnatggccccgaccgtggaaca
```
RandRBS1_luci_rv
```
Taactctagtactctagaag
```
RandRBS2_luci_fw
```
Rrrrrrrnnnnnnatggagaacatggagaacga
```
RandRBS2_luci_rv
```
Taaggatccttattacagct
```
J23113_carotene_fw
```
Tacggctagctcagtcctaggtatagtgctagcgcaagggcccaag
```
J23113_carotene_rv
```
Ttcacagagtggcctcgtga
```
crtE_mut_fw
```
gccgctatgccctgcatggacg
```
crtE_mut_rv
```
cgcggcttcgctgatcctt
```

**Code availability.** Code used in this study for model simulations was provided as Supplementary Material in Ceroni et al.[4]

**Data availability**. All data generated and analyzed during this study are provided with the published article as Supplementary Data 1. All other data are available from the authors upon reasonable request.

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

## Acknowledgements

O.B., C.B., M.M. and B.R.-M. were supported by the EPSRC grant EP/M002306/1. G.-B.S. was supported by the EPSRC fellowship EP/M002187/1 and grant EP/P009352/1. T.E. was supported by the EPSRC fellowship EP/M002306/1 and grant EP/J021849/1. We thank Francesca Ceroni for critical reading of the manuscript.

## Author contributions

O.B., C.B. and T.E. conceived and designed the experiments. O.B., C.B., M.M. and B.R.-M. performed the experiments. O.B. and G.-B.S. performed the simulations. O.B., G.-B.S. and T.E. analysed data. O.B., G.-B.S. and T.E. wrote the paper.

## Additional information

**Competing interests:** The authors declare no competing interests.

