## [Peer Review File · Nature Communications]

REVIEWERS' COMMENTS:

Reviewer #1 (Remarks to the Author):

The authors have now addressed my concerns. I hope the circularity issue I raised at first is now resolved. I believe the paper is now publishable in nature communications

Reviewer #2 (Remarks to the Author):

The author's rebuttal did not adequately address using the in vitro prediction of resource burden to forward design a metabolic pathway or synthetic circuit. For example, the authors could demonstrate that the in vitro method to predict resource burden could be used to reduce the frequencies of evolutionary escape of a synthetic circuit in vivo by improving the balance between target function and cellular fitness. I was not fully satisfied with the authors response about designing an optimal metabolic pathway since it seems that this method could be used to improve product yield for a metabolic pathway that is limited by resource burden. Without a clear demonstration illustrating how the in vitro method could be applied to improve synthetic circuit design, the paper may be a better fit for a focused journal such as ACS Synthetic Biology.

Reviewer #4 (Remarks to the Author):

I have now reviewed the manuscript and find that the authors adequately address the comments and requests from the different original reviewers. The manuscript provides an interesting and useful method for addressing the burden of protein production as well as the capacity to express proteins. The addition of data for pathways with multiple enzymes further demonstrates the usefulness of the method.

With regards to the comment from Reviewer 3 "Table S1: for each sequence it would be good to have the upstream and downstream sequences to have a better idea of the cloning and the whole regulatory part.", I do agree that extended sequences could be provided in the manuscript (and not only upon submission of plasmids to Addgene).

We wish to thank all reviewers for their continued feedback on our manuscript. Below are point-by-point replies to this second round of reviews.

Reviewers Comments

Referee #1 (Remarks to the Author):

The authors have now addressed my concerns. I hope the circularity issue I raised at first is now resolved. I believe the paper is now publishable in nature communications

We thank the reviewer for reading the revised manuscript and for their swift and positive decision.

Referee #2 (Remarks to the Author):

The author's rebuttal did not adequately address using the in vitro prediction of resource burden to forward design a metabolic pathway or synthetic circuit. For example, the authors could demonstrate that the in vitro method to predict resource burden could be used to reduce the frequencies of evolutionary escape of a synthetic circuit in vivo by improving the balance between target function and cellular fitness. I was not fully satisfied with the authors response about designing an optimal metabolic pathway since it seems that this method could be used to improve product yield for a metabolic pathway that is limited by resource burden. Without a clear demonstration illustrating how the in vitro method could be applied to improve synthetic circuit design, the paper may be a better fit for a focused journal such as ACS Synthetic Biology.

The forward design of a metabolic pathway or dynamic circuit is a typically a study in and of its own right. While such an effort would definitely add to the impact of our work here, we feel that this is beyond the scope of our work. Indeed, we would anticipate that such extra work would take several more months, especially if they incorporate passaging and growth rounds as required evolutionary studies. This extra time would unnecessarily delay the publication of our approach and prevents others from being able to immediately use this method in their work.

In the revised manuscript we have now been careful to not make claims that our approach will be able to optimise a pathway or dynamic circuit. The manuscript focus is on being able to predict expression costs for multigene systems and it is future work that will hopefully determine how this can be advantageous for pathway optimisation and circuit robustness and performance.

Referee #4 (Remarks to the Author):

I have now reviewed the manuscript and find that the authors adequately address the comments and requests from the different original reviewers. The manuscript provides an interesting and useful method for addressing the burden of protein production as well as the capacity to express proteins. The addition of data for pathways with multiple enzymes further demonstrates the usefulness of the method.

With regards to the comment from Reviewer 3 "Table S1: for each sequence it would be good to have the upstream and downstream sequences to have a better idea of the cloning and the whole regulatory part.", I do agree that extended sequences could be provided in the manuscript (and not only upon submission of plasmids to Addgene).

We thank the reviewer for reading the revised manuscript. We are glad that they feel positive about it.

We have now added the upstream and downstream sequences into Table S1 to provide better context for those interested in the DNA sequences used in this study.